# Coproduction Within Intersectoral Collaboration in the Context of a Neighborhood with Low Socioeconomic Scores in The Netherlands

**DOI:** 10.3390/ijerph22060954

**Published:** 2025-06-18

**Authors:** Roos van Lammeren, Jelmer Schalk, Suzan van der Pas, Jet Bussemaker

**Affiliations:** 1Department of Public Health and Primary Care, Health Campus the Hague, Leiden University Medical Centre, 2511 DP The Hague, The Netherlands; 2Faculty of Social Work and Applied Psychology, Leiden University of Applied Sciences, 2333 CK Leiden, The Netherlands; 3Institute of Public Administration, Leiden University, 2501 EE Den Haag, The Netherlands

**Keywords:** population health, coproduction, intersectoral collaboration, boundary spanner, neighborhood with low socioeconomic scores, action research

## Abstract

Intersectoral collaboration between health care, social care and other sectors has been widely advocated to improve population health outcomes. Similarly, the active role of citizens as coproducers is increasingly described in the literature as an important element for improving people’s health and well-being. Yet, there is little understanding of the role of coproduction in intersectoral collaboration, particularly in neighborhoods with low socioeconomic scores (SESs). In this empirical study, we analyze two aspects of coproduction that potentially drive positive health outcomes in intersectoral collaboration: How do coproducers in neighborhoods with low socioeconomic scores actively contribute to intersectoral collaboration, and what role does the relationship between professionals and citizens play in shaping these contributions? The study was conducted in a low-SES neighborhood in The Hague, the Netherlands. In this study, we explored the team ‘the Connectors’, an intersectoral collaboration of professionals and citizens with various (professional) backgrounds, focusing on accessible support for mental health services. The cause of mental health problems in low-SES neighborhoods varies; therefore, intersectoral collaboration is required in the approach to addressing these mental health problems. Using an action research approach, we demonstrated the importance of a reciprocal relationship between coproducers and professionals. We also found that ‘boundary spanners’ can help to improve this relationship, regardless of whether they are professionals or coproducers. We conclude that citizens in a low-SES neighborhood can not only benefit from coproduction, but can also contribute to it, because they have a high incentive to improve their neighborhood together with professionals in the intersectoral collaboration.

## 1. Introduction

Intersectoral collaboration, defined as ‘the recognized relationship between part or parts of the health sector with parts of another sector which has been formed to take action on an issue’ [1], has been widely advocated to improve population health outcomes [2]. Population health outcomes are determined by a wide range of factors, partly outside of the direct influence of the health care sector. The idea is that actively connecting professional expertise, information and resources from different sectors can achieve health outcomes ‘in a way that is more effective, efficient or sustainable than could be achieved by the health sector acting alone’ [1].

In the public administration literature, coproduction refers to the introduction of user-generated knowledge in joint public service delivery by individual citizens and professional providers [3]. In its most integrated form, coproduction constitutes a situation where clients are directly involved in both the design of the service provided to them and the production of this service by an organization, or a group of organizations [4]. Inferring from the public administration literature, we define coproduction as joint professional and citizen service delivery for and by the community.

Much of the work on coproduction in health care focuses on individual health outcomes resulting from some type of client involvement in the service provision by a health care organization, for example, the involvement of patients in a mental health care center to improve their own treatment plan [5]. However, this kind of focus excludes the community in which patients or clients reside. Yet, the community or neighborhood context is often where one can identify and leverage many of the resources that address the myriad of social, environmental and other factors affecting health outcomes, which are not directly related to health care. Moreover, improved health and health behaviors of individual clients can reciprocally benefit others in the neighborhood, for example, through improved education and social support.

Recent studies have highlighted the importance of coproduction in intersectoral collaboration to improve health outcomes [6,7,8,9]. Thus, coproduction within the community can contribute to intersectoral collaboration, at least in theory, as both focus on combining and integrating resources from the health sector and other sources for a specific population. Therefore, we define intersectoral collaboration as the collaboration between both professionals from diverse sectors and coproducers. Based on the existing literature, we suggest two aspects of coproduction in particular as driving positive health outcomes in intersectoral collaboration: (1) the active contribution of citizens as coproducers in intersectoral collaboration, and (2) the relationship between professionals and coproducers in this intersectoral collaboration.

First, the active contribution of citizens as coproducers of health services improves both the quality of and access to care and care coordination [10,11]. This is mainly because engaging clients in their own care promotes increased confidence and willingness to take control of their own health, resulting in better health and health behaviors, which in turn can inspire and help others. Further, coproduction contributes to intersectoral collaboration due to the citizens’ knowledge. These citizens know about the enablers and barriers in relation to effective (health) service implementation in their neighborhood context, which is, by definition, intersectoral. This knowledge comes from both their own and others’ previous exposure to services, which is why they have been termed ‘experts by experience’ [10].

Second, we are interested in the relationship between professionals and citizen coproducers. Previous research suggests that responsibilities cannot simply be shifted to citizens without professional support, in view of the considerable burden—not only emotional, but also physical and time-related—placed on the coproducing citizens [12]. In addition, the way professionals behave and interact with coproducers determines the success of coproduction [13,14]. Although the active involvement of citizens can increase the level of trust between citizens and professionals, the inherent power imbalance that exists between them needs to be addressed by the organizations of diverse sectors involved to create an inclusive, accessible and supportive collaboration [15]. Finally, the specific role that professionals adopt in their relationship with citizens, such as friend, leader, representative or mediator [16], affects the level of empowerment and the inclusion of vulnerable citizens. For example, the role of mediator or the so-called ‘boundary spanner’ [17,18] is important for bridging the information gap between professionals of various sectors and for connecting other citizens to the service provided by the coproduction [3,19,20]. The boundary spanner brings information from one group to another, and increases understanding and collaboration between groups [17,18].

The literature shows that coproduction can contribute to population health outcomes, but, as yet, it remains unclear precisely how coproduction can complement intersectoral collaboration, especially in low-SES neighborhoods. Building on these diverse findings, we explore how community-level coproduction contributes to intersectoral collaboration. Our research was conducted in the context of a neighborhood with low socioeconomic scores (SESs). We contribute to the literature, as there is a lack of evidence on how coproduction can be improved, particularly in neighborhoods with low SESs [21]. Low SESs are characterized by the following indicators: unemployment, low income, low education and low-paying jobs [22]. Research has shown that living in neighborhoods with low SESs increases the risk of poor general and mental health and individual health risk behavior [22,23,24]. Persuading citizens who live in a neighborhood with low SESs to participate in coproduction can be particularly challenging, due to the typically high level of distrust and low level of activism [25]; however, if successful, it may lead to comparatively high gains in terms of inclusion, empowerment and equity, and thus improved health outcomes [26].

The main research question of this study has two aspects: How do coproducers in neighborhoods with low socioeconomic scores actively contribute to intersectoral collaboration, and what role does the relationship between professionals and citizens play in shaping these contributions?

## 2. Materials and Methods

### 2.1. Study Setting

#### 2.1.1. Background

Our research context is The Hague, the third-largest city in the Netherlands. More specifically, this study was conducted in Moerwijk, a neighborhood of The Hague with low SESs [27] and a high percentage of citizens with a migrant background; specifically, this was 79.8% in 2024 [28]. The healthy life expectancy of men and women in neighborhoods with low SESs compared with other neighborhoods in The Hague shows a difference of 11 years and 13 years, respectively [27]. The local government is therefore working with health and welfare organizations and health insurers to make additional investments in neighborhoods with low SESs to equalize them in relation to other neighborhoods. This collaboration is a joint commitment called Healthy and Happy The Hague (HHTH), which focuses on both joint service delivery and pilot projects to improve the health of citizens in neighborhoods with low SESs [29]. In addition, in this neighborhood, there is generally a low trust in governmental organizations due to historical experiences such as the ‘childcare allowances affair’ in the Netherlands, which came to light in 2018 and highly affected families in this neighborhood. In this affair, dual nationality and low income turned out to be major risk factors for being wrongly designated as a fraudster; as a result, the affected families were pushed into poverty and placed under family control [30].

In collaboration with citizens of Moerwijk, HHTH developed the ‘Moerwijk social contract’ as one of its projects. The aim of this project is to better tailor care and support to the needs of (potential) service users. The project focuses on different ways of organizing, monitoring and financing care and support. The basic principles are that the project is guided by the neighborhood’s needs and that citizen coproducers, professionals and policy makers shape a new way of working in coproduction, based on experiences in other neighborhoods and countries, such as Birmingham, United Kingdom [31].

#### 2.1.2. The Connectors

The current study focuses on one intervention within the ‘Moerwijk social contract’ project, concentrating on accessible mental health support in the neighborhood. In September 2022, a team named ‘the Connectors’ was launched (in Dutch: de Verbinders). This is an interdisciplinary team consisting of two project leaders—a community builder from the neighborhood and a professional working with HHTH—together with two professionals (a psychotherapist and a social worker) and a citizen as coproducers from the neighborhood (Table 1). All team members hold paid positions. This team, the Connectors, works on specific requests for help raised by citizens in the neighborhood. These could be collective requests from several neighbors, such as how to deal with a mentally disturbed man in the neighborhood who can become aggressive; however, often, they are individual requests, for example, preventing an imminent eviction due to nuisance caused by a substance-abusing ex-partner who regularly shows up on the doorstep unexpectedly. This team of Connectors is easily accessible for citizens because of their active presence in community centers and non-reliance on fixed protocols, intentionally leaving room for tailor-made, intersectoral care solutions. The team aims to collaborate with citizens and active citizens as coproducers from the neighborhood, but also focuses on intersectoral collaboration by collaborating with a range of service providers, including mental health care, primary care, welfare and social care (Table 2). The overarching goal is to bring the network of citizen coproducers and professionals closer together in a way that allows them to complement one another when providing support in the neighborhood.

The composition of the team changed over time, with the social worker leaving in October 2023 and the coproducer in December 2023. In 2024, a number of people were added to the team (Table 1): a mental health nurse, an outpatient supervisor with a social work background, a community worker and a project employee who focuses on practical matters and preconditions for the team to function (such as external communication and organizing the financial budgets needed for various cases). There was no direct replacement for the coproducer, but the first three new employees live in or near the neighborhood and know the area well, through previous work and life experiences. From their role as citizens in Moerwijk, they are able to initiate collaborations with other citizens and closely involve coproducers in the neighborhood with the team. These changes in the team composition did not influence the data collection process, i.e., no data points are missing. In fact, the feedback from the research findings and moments of reflection with the team actually helped transfer to new team members because it gave them an overview of the process.

Before the study started, all participants (the members of the Connectors team) received a letter informing them about the study. It was stated that they could, at any time, refuse to be involved in the study. This was repeated orally before the data collection started. All participants provided written informed consent for the data collection during the whole period of the research. It was obtained by the corresponding author at the first team meeting, before the data collection started. The team members who started in 2024 provided written informed consent at the first meeting with the corresponding author.

The caseload handled by the team can be divided into six types of problems: (1) mental health, (2) financial stress, (3) substance abuse, (4) violence, (5) stress regarding housing, and (6) problems with or relating to children. In almost all of the cases, a combination of at least two types of problem were present. The team handles fifteen to twenty cases at a time, with an average duration of three weeks. After this period, the problems may have been solved, or the case is transferred to the appropriate expertise institutions or care providers, collaborating with other professionals needed to resolve it. In a third of the cases, the psychotherapist in the team can provide mental health treatment directly, for example, with trauma therapy. This treatment takes two to three months. In addition, each day, the team talks to roughly twenty-five people, to give support or to answer and refer short questions. To provide low-threshold assistance to citizens, the team does not immediately record the names and details of those involved.

### 2.2. Study Design

We used action research as the study approach for this research. Both the researchers and the practitioners in this study of the Connectors are acting together in the action research and focus on change and reflection. The iterative process of action research holds a cycle of activities, including problem diagnosis, action intervention and reflective learning [32]. This made it possible to both evaluate and develop the Connectors initiative within the study, giving us more insight into the barriers and enablers that influence the effectiveness of coproduction in intersectoral collaboration in neighborhoods with low SES. Reflection on the development process of the Connectors took place during the study with both the team members and project leaders. Within the action research, six different elements were used, as shown on the timeline in Figure 1: (1) observations, kept in logs, (2) evaluation interviews, (3) the Coordinated Action Checklist (CAC), (4) informal coffee moments with citizens, (5) reflection sessions with the team members, and (6) group sessions with the team members and project leaders. All these methods are described below.

Our method builds upon and extends the study of De Jong et al. [33]. Their approach effectively measured the collaboration within a coalition and mapped the network of those coalition members. In our study, we added further data collection elements to the method used by De Jong et al. [33], namely observations of the team meetings (Element 1), coffee moments with citizens (Element 4) and reflection sessions with the team members (Element 5).

The instrument used, the Coordinated Action Checklist (CAC), will be discussed in more detail below. We will start by looking at the other elements of data collection. First, the team meetings were observed and kept in logs for three months using the ‘Complete observer’ method [34] by the corresponding author (Element 1, Figure 1). These observations and its logs contributed to the aspect of the research concerning the relationship between professionals and citizens, by providing insight into the functioning and development of the team’s working method. After one year, individual interviews were held with the three executive team members (Element 2, Figure 1). An audio recording was made for all individual interviews. These interviews were conducted to gain more insight into both the active contribution of coproducers and the relationship between professionals and citizens, by asking how team members experienced the barriers and enablers of the first year. The interviews were semi-structured, evaluating members’ contributions within the team and the maturation of the team and external relations. Next, informal coffee moments with citizens in three different community centers took place (Element 4, Figure 1). The aim was to gather information on citizens’ opinions about developments in the neighborhood, including the introduction of the Connectors. Typically, only citizen coproducers attended organized meetings, and citizens said that they did not always feel free to speak openly about every topic during organized meetings. We therefore tried to speak with citizens in the informal setting of coffee tables in the community centers. The reflection sessions with the team (Element 5, Figure 1) were aimed at reflecting on the team’s development, and provided information on both the active contribution of citizen coproducers and the relationship between professionals and citizens. After the first year, the team informed us that the reflective topics were no longer feasible in the normal weekly team meetings, due to the pressure of the caseload. The team did have the need for reflection on the process; therefore, separate sessions were organized to continue learning as a team.

All collected data were used as input for the three group sessions in order to learn from the insights gained from these data, thus elucidating both the active contribution of coproducers and the relationship between professionals and citizens (Element 6, Figure 1). An audio recording was made of all group sessions, and they were moderated by the first author. The focus group methodology is used for the group sessions in order to gather information and reflect with the team in an effective way [35]. This method also gives the opportunity for interaction that reveals everyone’s perspective beliefs and values on a certain topic [35]. The whole team participated in the group sessions: both the team members and the project leaders. The purpose of the first group session was to evaluate the first year and set long-term goals for the team. The second group session covered the obstacles experienced in the first year and considered how to overcome them in the future, as discussed in the first group session. The goal of the third group session was to evaluate the visualization of the team’s network and the collaboration with those network partners. Social network analysis was performed, but it is too in-depth for this paper. Anyone interested may request the analysis from the corresponding author.

#### Coordinated Action Checklist (CAC)

The Coordinated Action Checklist (CAC) is a questionnaire used to discuss and evaluate the collaborations between stakeholders within local intersectoral collaboration [36]. It generates actionable knowledge for the partnership and therefore gives more insight into how the intersectoral collaboration operates [36]. The CAC was sent to all three team members and the two project leaders in September 2023 (Element 3, Figure 1). After everyone filled out the questionnaire, the scores were calculated per statement and per dimension into which the statements are divided [33]. The six dimensions consist of twenty-six statements in total, with a five-point scale from totally disagree to totally agree (see Appendix A). These dimensions were also found to be important in other studies where participation and cooperation were assessed [37]. The CAC is intended as a conversation starter and should be followed up in a group discussion [36], which was carried out in the second group session. Due to time and resource constraints, the CAC was not repeated thereafter. However, the topics derived from the CAC were discussed and evaluated again at the various reflection sessions.

### 2.3. Analytical Strategy

The overall data analysis consisted of a combination of verbatim transcriptions of the evaluation interviews, the CAC, the notes from the group sessions and observed team meetings, and the logs kept during the whole period of the project. All qualitative data were imported to ATLAS.ti 22 for analysis. A deductive, thematic analysis, as described by Pearse [38], was conducted, guided by the topics derived from the CAC. Additional codes were created if other relevant topics appeared from the data. Table 3 shows an overview of the final topics used in the analysis process. All authors contributed to the conceptualization and design of the study. The corresponding author led the data collection and analysis process. All four authors were involved in the final interpretation of the data.

## 3. Results

In this section, the development of the complex process of initiating and capitalizing on coproduction within intersectoral collaboration will be presented. We will start by elaborating the findings from the CAC because it reflects the team’s overall learning process. Next, findings relating to the first aspect of the research question, namely the active contribution of the coproducers, are described. We will then present the findings for the second aspect: the relationship between professionals and citizens. Finally, we describe our findings in the context of a neighborhood with low SESs and its influence on the study.

The Coordinated Action Checklist (CAC), Appendix A, shows how the Connectors value the relations in the team and towards their collaborative partners. The scores show the mean score of the individual ratings for each subject from 1 to 100. In itself, these evaluations are not valid measurements of the underlying concepts [36]. However, they are the starting point of the joint evaluation of the collaborations in the network, since the CAC is intended as a conversation starter [36]. The second group session (see Figure 1) was dedicated to the discussion following the CAC. We paid attention to the themes which received high scores (i.e., the growth dimension and the suitability of partners) to learn why these themes have higher scores than the other themes, and to find out how to maintain this. In addition, we dwelt on the other themes and what it would take for these to be scored higher. Individual questions from the checklist that scored remarkably low were likewise highlighted. For example, the fourth question “The contribution of the different members is to everyone’s full satisfaction” received low scores compared to other questions. It appeared that contributions from team members were not always acknowledged by other team members. This was also evident from results obtained from the individual evaluation interviews and other group sessions. The discussion showed us that for them, the contribution statement is closely aligned with the 11th statement “The partnership functions well (working structure, working methods)” and the 15th statement “The partnership partners work together in a constructive manner and know how to involve each other when action is needed”. Since the team started without clear guidelines and figured out by themselves what works and what does not, it appeared that the team members had different ideas of what contributes to the team goals and what does not. The individual evaluation interviews showed similar findings. Due to insufficient evaluation of the process in the first year (statement 12), which was also a topic of discussion in the first group session, this group session revealed that agreement on the goals of the team (statement 9) has to be discussed and improved. This was followed up in the following weeks and observed in our fieldwork during that period. From this discussion about the statements in the CAC, the team agreed to improve communication and express expectations to each other. Moreover, the reoccurring reflection sessions started as a result of the second group session about the CAC. The questionnaire was not repeated at a later time, but the topics of discussion were perceived as a valuable addition to the team’s evaluation and learning process and did return in the various reflection sessions. For this reason, the topics of the CAC are also integrated in other data sources.

### 3.1. Coproduction Within Intersectoral Collaboration—The Active Contribution of Coproducers

The first aspect we identified was the important contribution of the citizen coproducers. The data revealed three facets of the citizens’ active contribution within intersectoral collaboration.

First, coproducers serve as a bridge between citizens in need of support or care and professionals of the diverse organizations that provide the right assistance. It was observed in the team meetings (Element 1) and discussed in the first group session (Element 6) and several reflection sessions (Element 5) that trust between citizens and professionals is crucial in coproduction, otherwise the coproducers will not be able to fulfill the bridging function between the other citizens and professionals. The coproducers ask the Connectors for advice in certain cases, or for information on where they should refer a specific citizen in need. This was evident from the third group session, where the teams network and the collaboration within the network was discussed (Element 6), alongside the coffee moments with citizens (Element 4). The coproducers and other citizens spoke about the importance of the Connectors as easily accessible professionals, who can be asked all kinds of (complex) questions.

Second, the logs (Element 1) and reflection sessions (Element 5) indicated that the coproducers are aware of what happens in the neighborhood, which is not always the case for professionals. The coproducers live and participate in Moerwijk, and therefore have substantial knowledge about the neighborhood, whereas the professionals are only in the neighborhood during office times. The coproducers are (or were) also citizens in a vulnerable situation themselves and can therefore relate to situations from their own experience and act quickly when needed: they are experts by experience.

The third facet of the coproducers is the aim of communicating the voices of the highly diverse group of citizens and their needs in terms of support, in as representative a manner as possible (Quote 1). In general, citizens from neighborhoods with low SESs are categorized as hard to reach, and are therefore underrepresented at consultation meetings [26]. Coproducers can fulfill the role of a representative of the other citizens with all the knowledge they have about the neighborhood.

Quote 1—Evaluation interview (Element 2), team member:

“*I now participate much more in all kinds of meetings and networks where it is necessary to introduce the informal voice and the importance of our society. Also to be able to speak about what an informal network needs*.”

There is a downside to the three facets of the coproducers’ contributions which came to light in the evaluation interviews (Element 2), the CAC (Element 3) and the first two group sessions (Element 6) where the previous elements were discussed. All team members addressed the high amounts of pressure on the coproducers, and how they are constantly asked questions (Quote 2). Professionals who collaborate with these coproducers also tend to ask too much of just a few of them, without regard to their skills and capabilities. One example is that coproducers are asked to keep an eye on citizens who have been discharged from rehabilitation or psychiatric clinics. This is feasible when the professionals can be contacted if the coproducers find that professional help is needed. However, there were several known instances where the Connectors had to intervene in order to re-engage the responsible professionals because the coproducers themselves were unable to contact them.

Quote 2—Evaluation interview (Element 2), team member:

“*They [coproducers] work 36 hours a week as an (informal) caregiver/service provider/community center coordinator in the neighborhood, and in the evenings in the supermarket they are also approached about all kinds of things. That is quite difficult, of course. That will also be the case for the team member of the Connectors, who lives in the neighborhood and has no formal caregiver experience. There is no button to turn off. You are busy for 120 hours a week, or even longer*.”

As can be seen from Quote 2, the coproducer in the Connectors team has also experienced high amounts of pressure. Her position in the team is also vulnerable if she is expected to provide citizens with assistance for which she has not been trained. This was evident after a delay occurred in beginning to provide support for a complex case, and this coproducer was confronted with the situation in the evenings and weekends, outside of working hours. The risk of placing responsibility for complex situations on coproducers is that they can be confronted with the situation at any time. Coproducers have no time off from the neighborhood and its complex cases. This also played a role in the coproducer’s decision to stop being part of the team of Connectors.

Established on this downside, the team concluded that the coproducers need the support of professionals, and training in the tasks they fulfill in the neighborhood, provided by professionals (Quote 3). As a team member of the Connectors, coproducers can be of added value as trusted neighbors and in connecting with other citizens, as well as informing the other team members about their knowledge of the neighborhood, for instance, about tensions and peculiarities in the neighborhood. The tasks for the coproducer should be demarcated to their expertise. We cannot expect a coproducer to provide care for which he or she is not trained. In the first phase of the Connectors, too little attention was paid to the vulnerable position of the coproducer. Due to the reflective learning of the team this changed over time.

Quote 3—Reflection session (Element 5), team member:

“*Support is needed from a citizens’ network that takes on a formal role in the neighborhood. They are all people with certain baggage or a difficult past, who want to participate but need support. They are doing quite well professionally, but when things get difficult, personal problems become visible*.”

In addition, coproducers and team members stated that monetary support for coproducers helps them to feel they are taken seriously and shows appreciation for their important contribution (Elements 4 and 6). Career opportunities can also contribute to the citizens’ sense of dignity. Based on the evaluation interviews (Element 2) and the group session followed by these interviews (Element 6), we state that the context of a neighborhood with low SESs is an important factor in the type of recognition: these citizens, often in a vulnerable situation themselves, benefit from extra monetary and career support, as this reduces stress in their own (financial) situation.

### 3.2. Coproduction Within Intersectoral Collaboration—The Relationship Between Professionals and Citizens

The second aspect of coproduction within intersectoral collaboration concerns the relationship between professionals and citizens. We will introduce three steps that were identified in arriving at the reciprocal relationship between citizens and professionals that is required for coproduction. We found two obstacles for professionals participating in intersectoral collaboration and collaborating with coproducers. We will also introduce the team members fulfilling the role of ‘boundary spanners’, with three facets found to build and maintain relationships between professionals and citizens.

#### 3.2.1. Reciprocal Relationship Between Coproducers and Professionals

For coproduction within intersectoral collaboration to succeed, there must be a reciprocal, or equal relationship without power imbalances between professionals and coproducers [15]. From the logs (Element 1), the discussion followed by the CAC (Element 3), the reflection sessions (Element 5) and group sessions (Element 6), we identified three steps in reducing power imbalance and reaching a more balanced and reciprocal relationship, which align with the action-oriented guiding principles for achieving successful community engagement identified by De Weger et al. [15]. The first step towards achieving reciprocity in coproduction is for professionals to show courage and willingness to collaborate with coproducers. We saw professionals leave their familiar work environments to actively engage with the neighborhood and collaborate with coproducers. Second, reciprocity requires that the professionals trust the coproducers, and the coproducers trust the professionals. Generally speaking, citizens in Moerwijk distrust governmental organizations due to negative experiences with institutions in the past, such as the ‘childcare allowances affair’ that mainly affected low-SES neighborhoods. This is also one of the reasons why citizens see recording of their data as an obstacle to asking for help. Time and attention are needed to reduce this distrust. With the Connectors, we observed that returning weekly to all the different meeting places instead of a one-time visit helped build trust between the professionals and coproducers. Third, the Connectors found that being transparent about their work and all the collaborating partners helps to create trust in citizens and coproducers. The team is not choosing the side of public institutions by collaborating with organizations, but is rather trying to involve all the organizations needed to improve the support provided to citizens. We observed that being mutually transparent about the processes and any potential obstacles in (arranging) collaborations contributes to building trust between professionals and coproducers and forming a reciprocal relationship.

#### 3.2.2. Obstacles for Professionals to Collaborate

We found two obstacles for professionals participating in intersectoral collaboration and collaborating with coproducers in the evaluation interviews and group sessions (Elements 2 and 6). The first obstacle is the prior history of collaboration, which sets the initial trust level and therefore influences the collaborative process. Professionals operating in the neighborhood experienced failures in earlier short-term projects aimed at building a bridge between various organizations and coproducers, in order to improve the care for the people with complex cases (Quote 4). As a result, the professionals did not trust the Connectors in the initial phase of the project. This trust was slowly built as the team showed that the Connectors initiative is not a short-term project.

Quote 4—Group session 1 (Element 6), team member:

“*And in the formal network there you really notice the trauma of having a new organization and they are going to try something new. ‘We have already had ten of you “connectors”, you know… So just tell us exactly what you’re going to do.’ And even if you explain it in the best way, it is still not trusted. ‘Let’s wait and see’ is the attitude of most professionals*…”

The second obstacle experienced in practice concerns the constraint on participation due to the pressure of rules and regulations. From the reflections on the collaboration in both the evaluation interviews of the team members (Element 2) and the group sessions (Element 6), it was evident that rules and regulations, such as the General Data Protection Regulation (GDPR), feel like obstacles to professionals, and prevent them from participating. This burden is imposed by the organization to which those professionals belong. Some organizations and their professionals report that rules like the GDPR do not have to be an obstacle to participating in collaboration. The team argues that the organizations and their professionals who feel the pressure of rules and protocols should instead prioritize care for citizens in need (Quote 5). This result shows that there is a need for support and dialog about how to overcome the obstacle of regulations.

Quote 5—Group session 3 (Element 6), team member:

“*Unfortunately, some organizations still reason and act based on the rules first, which means that assistance for citizens has lower priority*.”

These observed obstacles mean that in practice, collaboration depends on several enthusiastic professionals who dare to fully commit to the collaboration promoting care of the citizens, which makes it vulnerable, as indicated in Quote 6. When these professionals leave, collaboration must be rebuilt with colleagues from the same organization. Creating support within the organization means that the vulnerability of the collaboration decreases.

Quote 6—Log of event with new and existing partners (Element 1), team member:

“*It is important that these individuals also transfer this to the rest of the organization so that the ‘oil slick’ can grow. If that does not happen and the individuals leave the organization, a piece of the network disappears and cooperation must be built anew.*”

#### 3.2.3. The Team Members as ‘Boundary Spanners’

We distinguished three facets of the Connectors team in fulfilling the role of ‘boundary spanners’ in the neighborhood when building and maintaining the relationship between professionals and coproducers. First, the team’s personal approach in assisting with a complex case was evident in the way the team proceeds when receiving new cases (Quote 7). The team identifies the context of the case and determines which professional(s) of the team, each with their own expertise and working method, fit(s) best with the case.

Quote 7—Evaluation interview (Element 2), team member:

“*So if there were individual problems, the psychotherapist of the team often tackled them. If something was group-oriented and system-oriented, the coproducer team member and I would focus on it. Because that is our background and that is what we are good at. Not that you completely let go of the individual situations, that is not possible, of course*.”

Second, from the logs (Element 1) and reflection sessions (Element 5), we determined that the team members assume a variety of roles within coproduction: leader, practitioner, friend, representative or mediator [16]. The team members led meetings in the neighborhood, acted as practitioners for citizens in need, but also took on the role of a friend that citizens can consult informally. There are further examples where team members represented a citizen (or group of citizens) in meetings with professionals. They also assumed the mediator role in conflicts between different coproducers or organizations within the collaboration. This corresponds to the role of a ‘boundary spanner’, who bridges the gap between professionals of diverse sectors and coproducers.

The third facet of the Connectors’ role as ‘boundary spanners’ that we observed is providing support at different levels. The varied backgrounds of the team members enable them to offer help to the neighborhood at individual, group-oriented and system-oriented levels (Quote 7). The individual level is achieved through traditional assistance and support for a single citizen. Examples of group-oriented support are sessions on street intimidation organized for young people, and grief counselling at the community center after the death of a well-known person from the neighborhood. The support at the system level is evident in the case of recurring problems that could be prevented: the team engages with the organizations in an attempt to initiate structural improvements in the support and care provided.

These results show that the Connectors play a crucial role in the relationship between professionals and citizens, by helping to increase trust in each other, facilitating contacts and encouraging collaboration between these groups. They promote the process of both intersectoral collaboration and coproduction; therefore, we identify them as ‘boundary spanners’ [18].

### 3.3. The Context of Neighborhoods with Low Socioeconomic Scores

The ability to fulfill the team’s role in a neighborhood with low SESs requires certain preconditions. This was evident from the evaluation interviews (Element 2) and the group sessions (Element 6) evaluating the CAC (Element 3), where the team members raised the topic of preconditions. Following their experiences in practice, they identified three preconditions to improve the working environment and discussed them with the project leaders. First, an emergency fund (in Dutch: handgeld) is needed for crisis situations of vulnerable citizens if there is no time to apply for money via the formal route (Quote 8). Second, a safe environment with a confidential counsellor is required, due to the severity and complexity of the caseload handled by the team. This is always important, but it was especially emphasized by the team members because of the context of a neighborhood with low SESs and a heavy caseload. Third, it is desirable to have a workspace with consultation and treatment rooms in the neighborhood, close to the citizens in need, for consultation with partners in the collaboration and to provide the necessary help to those citizens. Typically, social workers in the Netherlands visit citizens in need at home, but the team explained that this is not always feasible, in view of the housing situation in neighborhoods with low SESs.

Quote 8—Evaluation interview (Element 2), team member:

“*So, an emergency fund is also being looked at, but even for that we are now thinking on a large scale. Shouldn’t you have some kind of fund for the neighborhood from which you occasionally take 25 euros to help someone out of problems, without having to be subject to the Financial Supervision Act*?”

In addition to working in a neighborhood with low SESs, the coproducers are also citizens in a vulnerable situation themselves. The situations they have to deal with in that neighborhood are difficult and the coproducers have not had the necessary training. They therefore need support in fulfilling their role as coproducers. This raises the question of whether it is responsible to ask a coproducer to fulfill the role of ‘citizen professional’. This relates to the downside of coproducers’ contributions in the neighborhood, as discussed above, specifically the high amount of pressure and not being able to leave the neighborhood behind after a working day, in combination with insufficient training in how to handle this.

The vulnerability of neighborhoods with low SESs is reflected in the patience and time needed to build a good relationship of trust. The neighborhood is also characterized by different cultures and languages, which need to be considered when setting up intersectoral collaboration, including coproduction (Quote 9). The role of the Connectors is important, especially for incorporating coproduction within intersectoral collaboration. They do this by giving attention to the neighborhood’s characteristics and particularities, and have shown that they are patient and take the time to (re)build trust.

Quote 9—Group session 1 (Element 6), team member:

“*You have to understand the DNA of the neighborhood to know what you are talking about*.”

## 4. Discussion

This study analyzed the role of coproduction in intersectoral collaboration with respect to (1) the active contribution of citizens as coproducers in intersectoral collaboration, and (2) the relationship between professionals and coproducers in this collaboration. Both aspects of coproduction in intersectoral collaboration were studied in the context of a neighborhood with low socioeconomic scores.

The results relating to the first aspect showed that coproducers are the eyes and ears of the neighborhood and aim to communicate the voices of the highly diverse group of citizens in the neighborhood in as representative a manner as possible. Coproducers serve as a bridge between professionals and the population in need of support or care and the organizations that provide the right assistance. In line with the literature on intersectoral collaboration and coproduction, trust and trust building appeared to be crucial to connecting citizens to the right service provider [3,19]. In line with the findings of Van Eijk et al. [12], we observed that help is needed for coproducers. We add our findings to the existing literature, namely that the high amount of pressure on coproducers, especially in a neighborhood with low SESs, necessitates more support in the form of training and recognition. Which specific types of training and (monetary) recognition would be most effective for coproducers in a low-SES neighborhood is an interesting question for follow-up studies.

The second aspect showed the importance of a reciprocal relationship between coproducers and professionals. The obstacles for professionals to collaborate confirmed the existing literature of Ansell and Gash [39] on collaborative governance, and particularly the prior history of collaboration determining the initial trust level. This is in line with the more recent findings of Alderwick et al. [40], who identify trust as a basis for relationships and communication between professionals of health care and non-health-care organizations. We found that obtaining reciprocity is an unruly process, requiring a diversity of roles from the team members. This is in agreement with the literature that states the importance of fulfilling various roles to include and empower vulnerable citizens [16]. We add two other facets of ‘boundary spanners’ to our understanding of the role of coproduction in intersectoral collaboration: (1) the personal approach of the team members in assisting with a complex case and (2) the team members providing support to the neighborhood at different levels (individual, group-oriented and system-oriented). Our study also shows the importance of meeting three preconditions for the intersectoral team of professionals to perform their work: (1) an emergency fund, (2) a safe environment with a confidential counsellor and (3) a workspace with consultation rooms in the neighborhood. Follow-up research could study actionable strategies for implementing and sustaining these preconditions. We conclude that the role of a boundary spanner can be fulfilled by both coproducers and professionals. Previous research [41,42] shows that it is hard to develop a specific set of role expectations and competences for the boundary spanner role and therefore institutionalize it. The main reason is the context-dependency of this role. Our study found a considerable degree of variability in role behavior and expectations even across the few boundary spanners present in the local network. Future research should address whether and how such boundary-spanning roles can be institutionalized or structurally supported, reducing dependency on specific individuals to ensure sustainable and consistent community engagement.

Based upon our results, we conclude how crucial the context is—in this case, a neighborhood with low SESs—regarding attendance and difficulties. It determines the approach needed for successful coproduction within intersectoral collaboration. Previous research has shown that the difference in coproduction between neighborhoods is explained by their social capital [43]. A neighborhood’s spatial design is relevant [44] and may impact the formation of social capital [43]. This is in line with our results that show the important active contribution of coproducers to the neighborhood in building and maintaining social networks, thus improving its social capital.

There are additional challenges for coproduction in neighborhoods with low SESs. For example, with regard to each of the two aspects, we observed the crucial role of trust for both citizens and professionals, which is often lacking in the first stage of coproduction within intersectoral collaboration. As expected, we found the level of trust between citizens and professionals to be typically low in neighborhoods with low SESs, due to negative previous experiences with governmental organizations. Therefore, building trust is essential and requires extra attention in developing coproduction in these neighborhoods. Personal relationships, preferably long-term ones, are crucial for building trust. In general, it is essential to take the neighborhood context into account in terms of its level of deprivation when aiming for coproduction within intersectoral collaboration. This will also offer additional opportunities for the neighborhood, such as more sustainable and structural collaborations to address (mental) health problems of the citizens there.

This study focuses on the coproduction process of the Connectors team in Moerwijk, a neighborhood of The Hague. We used action research, which helped us to continuously reflect on practices and lessons, providing deeper insights into the development through participation in the whole process. We thus gained a better understanding of coproduction. This is, to our knowledge, the first study to look at the role of coproduction in intersectoral collaboration in a neighborhood with low SESs with respect to (1) the active contribution of the coproducers and (2) the relationship between professionals and citizens.

Our study emphasized internal validity through contextualized and triangulated data collection and process tracing. However, since a single (deprived) neighborhood was studied without a systematic comparison to other neighborhoods, it is hard to evaluate the generalizability of the findings. At the same time, it cannot be ruled out that the findings are in fact transferable to other contexts. Comparative neighborhood case studies are thus an important step for follow-up research”. Another limitation of this research is the small size of the Connectors team (five to seven people) and the high turnover in the team during the period covered by the study. On the one hand, this has an impact on the level of trust built up with other professionals and citizens in the coproduction in intersectoral collaboration. The longitudinal design of our study allowed us to observe a temporary drop in the level of trust directly following the departure of the coproducer and the social worker. In this particular network, the team was able to accommodate the transfer of tasks and relationships to the new hire rather efficiently, thus minimizing the negative impact of turnover on ongoing collaboration within the network overall. Yet, the single-case design precludes any evaluation of the robustness of this moderated effect of turnover across other, similarly deprived neighborhoods. On the other hand, new team members brought new energy and had new perspectives that contributed to the team. They also had their own networks, which expanded the team’s collaborative network.

The Connectors team and coproducers that collaborate within the neighborhood are aware of the diversity of community members, notably the high percentage of citizens with diverse migrant backgrounds. Even though they attempt to reach various groups in the community by collaborating with various coproducers and citizens, we are aware that some sub-groups may engage less effectively due to language barriers, cultural mistrust, or other socio-cultural factors. Future studies would benefit from explicitly investigating differential engagement levels and how coproduction interventions could be tailored more effectively to diverse community sub-groups, thereby enhancing equitable health outcomes.

## 5. Conclusions

The added value of this study is that it revealed detailed insights into how coproducers in neighborhoods with low socioeconomic scores actively contribute to intersectoral collaboration, and the role the relationship between professionals and citizens plays in shaping these contributions. Our research confirms that coproducers have an important role within the intersectoral collaboration in improving the health of citizens in the neighborhood [16,45]. We add that also citizens in a vulnerable situation are able to contribute and benefit from coproduction because they have a high incentive to improve the neighborhood together with the professionals of the diverse sectors. Coproducers have knowledge of the neighborhood and its problems, but they need support and training in how to handle difficult situations to improve their skills. There is a limit to how much can be asked of a coproducer within the intersectoral collaboration. This limit can only be extended with sufficient guidance and training of the coproducer. We conclude that especially in these neighborhoods, with low SESs and distrust towards institutions, the role of coproducers can be crucial in shaping necessary intersectoral collaboration.

## Figures and Tables

**Figure 1 ijerph-22-00954-f001:**
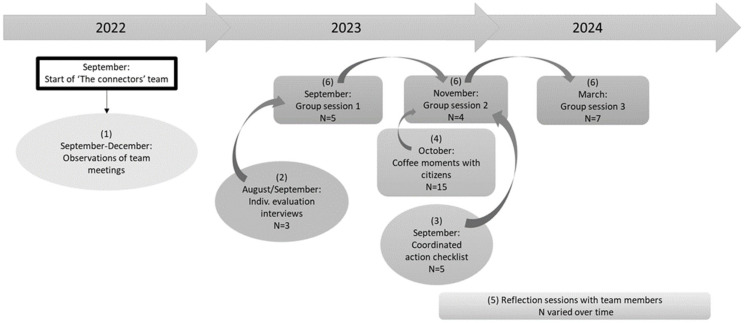
Timeline of research activities.

**Table 1 ijerph-22-00954-t001:** Composition of the Connectors team.

Composition of the Connectors Team Starting in 2022:	Composition of the Connectors Team from 2024 and Onwards:
Psychotherapist	Psychotherapist
Social worker	Mental health nurse
Active citizen as coproducer	Outpatient supervisor (social work background)
	Community worker
	Project employee

**Table 2 ijerph-22-00954-t002:** Overview stakeholders of the Connectors team.

Variety of Citizens in the Neighborhood Where the Connectors Collaborate with the Following:	Main Services in the Network of the Connectors:
Citizen coproducers, among others:	Mental health care
- Community center coordinators and volunteers.	Primary health care
- Food bank volunteers.	Welfare
- Grouped citizens organizing activities.	Social care
(potential) Service users.	Police
Citizens.	Housing corporation

**Table 3 ijerph-22-00954-t003:** Overview of the topics used in the analysis process.

Topics Derived from the CAC (Top Down):	New Topics that Emerged from the Data (Bottom Up):
Suitability of partners	Learning network
Task dimension	Professional–coproducer collaboration
Relation dimension	Experiences in practice
Growth dimension	
Visibility dimension	

## Data Availability

The data that support the findings of this study are available, but restrictions apply to the availability of these data. These data were used under license for the current study and so are not publicly available. The data are, however, available from the authors upon reasonable request.

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
