# Peer review of "Coproduction Within Intersectoral Collaboration in the Context of a Neighborhood with Low Socioeconomic Scores in The Netherlands"

_ijerph, 2025, doi:10.3390/ijerph22060954_

Round 1
Reviewer 1 Report
Comments and Suggestions for Authors
The manuscript by van Lammeren et al. provides a detailed examination of coproduction within intersectoral collaboration in a low socioeconomic status (SES) neighborhood in The Hague, Netherlands, through the lens of the "Connectors" initiative. The study’s action research approach, focusing on the active contributions of citizen coproducers and their relationships with professionals, is commendable for its depth and engagement with a vulnerable community. It contributes meaningfully to the literature by highlighting the role of "boundary spanners" and the contextual challenges of low SES settings. However, the study has several limitations that warrant critical scrutiny, including its scope, analytical depth, and generalizability. Below, I provide several evaluations and recommendations for improvement.
- The study’s single-case design, while rich in contextual detail, limits its broader applicability. The focus on Moerwijk, without comparison to other neighborhoods or initiatives, raises questions about whether the findings are unique to this context or transferable to other low SES settings. The authors acknowledge this limitation but do not sufficiently address how it impacts the robustness of their conclusions.
- The thematic analysis is well-structured, but the discussion of key concepts, such as the reciprocal relationship between professionals and coproducers, lacks depth. For instance, the three steps to achieve reciprocity (courage, trust, transparency) are presented without detailed exploration of how these are operationalized or sustained over time. Similarly, the concept of "boundary spanners" is compelling but underexplored in terms of how these roles can be institutionalized to reduce reliance on individual actors.
- The study notes the high percentage of citizens with migrant backgrounds in Moerwijk but does not adequately analyze how cultural or linguistic barriers affect coproduction. The claim that coproducers represent the "voices of all citizens" is ambitious and potentially overstated, given the acknowledged challenges of engaging hard-to-reach groups in low SES neighborhoods.
- While the study identifies preconditions (e.g., emergency funds, safe environments), it does not provide actionable strategies for implementing or sustaining these. The discussion of coproducer support (e.g., training, monetary recognition) is insightful but lacks specificity regarding what types of training or resources are most effective. This limits the study’s utility for practitioners seeking to replicate the Connectors model.
- The action research approach is appropriate, but the high turnover in the Connectors team (e.g., the departure of the coproducer and social worker) is underexplored as a methodological challenge. The impact of this turnover on data collection, trust-building, and the study’s findings is not sufficiently addressed. Additionally, the Coordinated Action Checklist (CAC) is a valuable tool, but its integration with other data sources could be better articulated to demonstrate triangulation.
The relevant sections should be re-evaluated and the discussion section should be strengthened, and if necessary, these limitations should be addressed.
Author Response
The manuscript by van Lammeren et al. provides a detailed examination of coproduction within intersectoral collaboration in a low socioeconomic status (SES) neighborhood in The Hague, Netherlands, through the lens of the "Connectors" initiative. The study’s action research approach, focusing on the active contributions of citizen coproducers and their relationships with professionals, is commendable for its depth and engagement with a vulnerable community. It contributes meaningfully to the literature by highlighting the role of "boundary spanners" and the contextual challenges of low SES settings. However, the study has several limitations that warrant critical scrutiny, including its scope, analytical depth, and generalizability. Below, I provide several evaluations and recommendations for improvement.
Comment 1: The study’s single-case design, while rich in contextual detail, limits its broader applicability. The focus on Moerwijk, without comparison to other neighborhoods or initiatives, raises questions about whether the findings are unique to this context or transferable to other low SES settings. The authors acknowledge this limitation but do not sufficiently address how it impacts the robustness of their conclusions.
Response 1: Thank you for your positive words and feedback on our paper. With respect to the single-case design, we agree that we need to elaborate further on its impact on the robustness of our conclusions. We have clarified this issue in the discussion section by addressing the internal and external validity of the study: “Our study emphasized internal validity through contextualized and triangulated data collection and process tracing. However, since a single (deprived) neighborhood was studied without a systematic comparison to other neighborhoods, it is hard to evaluate the generalizability of the findings. At the same time, it cannot be ruled out that the findings are in fact transferable to other contexts. Comparative neighborhood case studies are thus an important step for follow-up research.” (p.15, row 715-720).
Comment 2: The thematic analysis is well-structured, but the discussion of key concepts, such as the reciprocal relationship between professionals and coproducers, lacks depth. For instance, the three steps to achieve reciprocity (courage, trust, transparency) are presented without detailed exploration of how these are operationalized or sustained over time. Similarly, the concept of "boundary spanners" is compelling but underexplored in terms of how these roles can be institutionalized to reduce reliance on individual actors.
Response 2: Thank you for pointing this out. We further clarified the results regarding reciprocal relationship between professionals and coproducers by adding the following sentences, which aim to let the reader follow the steps to achieve reciprocity in the context of our case study:
- Courage: “We saw professionals leave their familiar work environments to actively engage with the neighborhood and collaborate with coproducers.” (p. 11, row 489-490)
- Trust: “With the Connectors we observed that returning weekly to all the different meeting places instead of a one-time visit, helped build trust between the professionals and coproducers.” (p. 11, row 496-498)
- Transparency: “We observed that being mutually transparent about the processes and any potential obstacles in (arranging) collaborations contributes to the trust building between professionals and coproducers and forming a reciprocal relationship.” (p. 11, row 502-505)
In addition, we agree that the concept of boundary spanners is compelling, as well as the question how this role can be institutionalized to reduce reliance on individual actors. We elaborate on this issue in the discussion (p. 14). We searched for references that could add to this topic and based on this search added the following sentence: “Previous research [41,42] shows that it is hard to develop a specific set of role expectations and competences for the boundary spanner role and therefore institutionalize it. The main reason is the context-dependency of this role. Our study found a considerable degree of variability in role behavior and expectations even across the few boundary spanners present in the local network.” (p. 14, row 671-675).
Comment 3: The study notes the high percentage of citizens with migrant backgrounds in Moerwijk but does not adequately analyze how cultural or linguistic barriers affect coproduction. The claim that coproducers represent the "voices of all citizens" is ambitious and potentially overstated, given the acknowledged challenges of engaging hard-to-reach groups in low SES neighborhoods.
Response 3: We agree that this statement is too ambitious and we nuanced the wording of the sentence: “aimed to communicate the voices of the highly diverse group of citizens in the neighborhood in as representative manner as possible” (p. 9 row 401-403 and p. 13/14, row 642-643).
Comment 4: While the study identifies preconditions (e.g., emergency funds, safe environments), it does not provide actionable strategies for implementing or sustaining these. The discussion of coproducer support (e.g., training, monetary recognition) is insightful but lacks specificity regarding what types of training or resources are most effective. This limits the study’s utility for practitioners seeking to replicate the Connectors model.
Response 4: Thank you for pointing out this interesting point. However, it is outside the scope of our research to develop detailed actionable strategies. This is certainly a relevant topic for follow-up research, which we now mention in the discussion: “The specific types of training and (monetary) recognition most effective for coproducers in a low SES neighborhood is an interesting question for follow-up studies.” (p.14, row 650-652) and “Follow-up research could study actionable strategies for implementing and sustaining these preconditions.” (p. 14, row 669-670)
Comment 5: The action research approach is appropriate, but the high turnover in the Connectors team (e.g., the departure of the coproducer and social worker) is underexplored as a methodological challenge. The impact of this turnover on data collection, trust-building, and the study’s findings is not sufficiently addressed. Additionally, the Coordinated Action Checklist (CAC) is a valuable tool, but its integration with other data sources could be better articulated to demonstrate triangulation.
Response 5: Thank you for this insightful feedback. First of all, we share your concern regarding the high turnover in the Connectors team. Therefore, we address it as a methodological limitation of the study. The high turnover did in fact negatively impact trust building. At the same time, the team was able to accommodate the transfer of tasks and relationships to the new hire rather efficiently, thus minimizing the negative impact of turnover on ongoing collaboration within the network overall. To further scrutinize this point, we added the following sentence to this limitation: “The longitudinal design of our study allowed us to observe a temporary drop in the level of trust directly following the departure of the coproducer and the social worker. In this particular network, the team was able to accommodate the transfer of tasks and relationships to the new hire rather efficiently, thus minimizing the negative impact of turnover on ongoing collaboration within the network overall. Yet, the single-case design precludes any evaluation of the robustness of this moderated effect of turnover across other, similarly deprived neighborhoods.” (p. 15, row 724-730).
Despite the turnover in the team, we followed the data collection process as planned, i.e. no data points are missing. Therefore, we added the following sentence to the method section: “Those changes in the team composition did not influence the data collection process, i.e. no data points are missing. In fact, the feedback of research findings and moments of reflection with the team actually helped transfer to new team members because it gave them an overview of the process the team was in.” (p. 4, row 198-202)
The second part of this point of criticism concerns the integration of the CAC with other data sources. We aimed to address the concerns of both the first and second reviewer (see point 3 and 4 ‘Regarding Results’ of Reviewer 2 below) regarding the CAC in one additional paragraph added to the results section where we elaborate further on the tool and its role in triangulation (p. 8, row 342-274):
The Coordinated Action Checklist (CAC), Appendix A, shows how the team the Connectors values the relations in the team and towards their other collaborative partners. The scores show the mean score of the individual ratings for each subject from 1-100. In itself, these evaluations are not valid measurements of the underlying concepts [36]. However, they are the starting point of the joint evaluation of the collaborations in the network, since the CAC is intended as a conversation starter [36]. The second group session (see Figure 1) was dedicated to the discussion following the CAC. We paid attention to the themes received high scores (i.e. the growth dimension and the suitability of partners), to learn why these themes have a higher score than the other themes and how to maintain this. In addition, we dwelt on the other themes and what it would take to score them higher. Individual questions from the checklist that scored remarkably low were likewise highlighted. For example, the fourth question “The contribution of the different members is to everyone’s full satisfaction” received low scores compared to other questions. It appeared that contributions from team members were not always acknowledged by other team members. This was also evident from results obtained from the individual evaluation interviews and other group sessions. The discussion showed us that for them, the contribution statement is closely aligned with the 11th statement: “The partnership functions well (working structure, working methods)” and the 15th statement: “The partnership partners work together in a constructive manner and know how to involve each other when action is needed”. Since the team started without a clear guideline and figured out themselves what works and what does not, it appeared that the team members had different ideas of what contributes to the team goals and what not. The individual evaluation interviews showed similar findings. Due to insufficient process evaluation in the first year (statement 12), which was also a topic of discussion in the first group session, this group session revealed that agreement on the goals of the team (statement 9) has to be discussed and improved. This was followed up in the following weeks and observed in our fieldwork during that period. From this discussion about the CAC statements, the team agreed to improve communication and express expectations to each other. Besides the reoccurring reflection sessions started as a result of the second group session about the CAC. The questionnaire was not repeated at a later time, but the topics of discussion were perceived as a valuable addition to the team’s evaluation and learning process and did return in the various reflection sessions. For this reason, the topics of the CAC are also integrated in other data sources.
The relevant sections should be re-evaluated and the discussion section should be strengthened, and if necessary, these limitations should be addressed.
Response: Thank you for this feedback. With the re-evaluations of the specific points above and complemented with the processed suggestions of the other reviewers we made changes and strengthened the discussion and elaborated the limitations.
Reviewer 2 Report
Comments and Suggestions for Authors
Dear Authors, congratulations on your work!
The paper aims to analyse two aspects of coproduction that potentially drive positive health outcomes in intersectoral collaboration: (1) the active contribution of citizens as coproducers in this collaboration and (2) the relationship between professionals and coproducers.
It is well-written, presents contributions to public health and demonstrates careful planning and stakeholder involvement. However, the objective in the abstract requires more precise articulation, and there are methodological concerns regarding the distinction between case study and action research approaches.
Regarding Methodology:
- The authors must differentiate between the case study and action research approaches, as they are conceptually distinct methodologies with different epistemological foundations.
- A theoretical framework for the case study approach is missing. Which author(s) informed the case study design? What specific type of case study was employed?
- How did the authors record the interviews and group sessions? Was it video/audio? It seems that the authors used focus groups (as group sessions) as a methodology, and if so, you need to state the authors to support that method.
- Did the authors plan to reapply the CAC after 2023? It could be interesting to understand if there's any change in collaboration, especially after action research.
Regarding Results:
- The thematic analysis process is not sufficiently detailed. Which categories/themes emerged from this analysis? The authors could develop a table with this thematic analysis
- Did new topics emerge during the analysis that were not anticipated in the initial research framework?
- The contributions from CAC to the discussion are not clearly articulated. How did this analytical approach enhance the findings?
- The CAC scoring interpretation is not adequately explained. What do the different scores represent, and how should readers understand their significance?
Regarding References: the authors could update their references as only 13 of 40 are from the last 5 years (and some from municipal/national reports).
Author Response
Dear Authors, congratulations on your work!
The paper aims to analyse two aspects of coproduction that potentially drive positive health outcomes in intersectoral collaboration: (1) the active contribution of citizens as coproducers in this collaboration and (2) the relationship between professionals and coproducers.
It is well-written, presents contributions to public health and demonstrates careful planning and stakeholder involvement. However, the objective in the abstract requires more precise articulation, and there are methodological concerns regarding the distinction between case study and action research approaches.
Response: Thank you for the complements and valuable feedback. We changed the objective in the abstract on p. 1 (row18-20). Below we give a point by point response to the rest of the feedback.
Regarding Methodology:
Comment 1: The authors must differentiate between the case study and action research approaches, as they are conceptually distinct methodologies with different epistemological foundations.
Response 1: We understand the confusion because twice there was a sentence moderately worded in the text trying to describe that we studied one case i.e. a single-case design, but it appeared that we are doing a case study. This study uses an action research approach, not a case study methodology. We modified the sentences that may have caused the misunderstanding:
- “We used action research as the study approach for this research. Both the researchers and the practitioners in this study of the Connectors are acting together in the action research and focus on change and reflection” (p. 5, row 230-231).
- “However, since a single (deprived) neighborhood was studied without a systematic comparison to other neighborhoods, it is hard to evaluate the generalizability of the findings.” (p. 15, row 716-718).
Comment 2: A theoretical framework for the case study approach is missing. Which author(s) informed the case study design? What specific type of case study was employed?
Response 2: We hope we have resolved this point with the response to the previous point. We do not conduct a case study, but instead adopt an action research approach.
Comment 3: How did the authors record the interviews and group sessions? Was it video/audio? It seems that the authors used focus groups (as group sessions) as a methodology, and if so, you need to state the authors to support that method.
Response 3: Thank you for raising this question. We made audio records during the interviews and group sessions. Indeed we used the methodology of focus groups for the group sessions. We clarified both points in the text:
- “An audio recording was made of all individual interviews.” (p. 6, row 265-266)
- “An audio record was made of all group sessions and they were moderated by the first author. The focus group methodology is used for the group sessions in order to gather information and reflect with the team in an effective way [35]. This method also gives the opportunity for interaction that reveals everyone’s perspective beliefs and values on a certain topic [35].” (p. 7, row 289-293)
Comment 4: Did the authors plan to reapply the CAC after 2023? It could be interesting to understand if there's any change in collaboration, especially after action research.
Response 4: Reapplying the CAC would likely be very insightful indeed. Ideally, we would have liked to repeat the CAC after one year, but the workload of the team in that same period, as well as resource constraints on the part of the research team precluded this step. We used the questionnaire – as the CAC was intended – as a conversation starter and eventually the topics of the CAC, discussed in group session 2, were also discussed in the various reflection sessions. In this way, we were able to discuss the topics of the CAC at multiple points in time, although without quantitative scores (see also response to R1 point 5 above). To clarify this point in the paper itself, we added “The CAC is intended as a conversation starter and should be followed up in a group discussion [36], which is done in the second group session. Due to time and resource constraints the CAC was not repeated thereafter. However the topics derived from the CAC were discussed and evaluated again at the various reflection sessions.” (p. 7, row 311-315)
Regarding Results:
Comment 1: The thematic analysis process is not sufficiently detailed. Which categories/themes emerged from this analysis? The authors could develop a table with this thematic analysis
Response 1: Thank you for pointing this out. As stated on p. 7 we followed the themes of the CAC and when necessary added extra topics. To clarify the exact topics in the paper we added a table as suggested on p. 7 (327-333), including both the topics derived from the CAC and the new topics emerged during the analysis:
|
Topics derived from the CAC (top-down): |
New topics that emerged from the data (bottom-up): |
|
Suitability of partners |
Learning network |
|
Task dimension |
Professional-coproducer collaboration |
|
Relation dimension |
Experiences in practice |
|
Growth dimension |
|
|
Visibility dimension |
|
Comment 2: Did new topics emerge during the analysis that were not anticipated in the initial research framework?
Response 2: Yes, new topics did emerge. As described in the response to the previous point, these new topics are discussed and also shown in Table 3, on p. 7 (row 327-333).
Comment 3: The contributions from CAC to the discussion are not clearly articulated. How did this analytical approach enhance the findings?
Response 3: Thank you for this feedback. We address the concerns of both the first and second reviewer regarding the CAC by including an additional paragraph to the results section where we elaborate further on the tool, see p. 8, row 342-374 (See also the response to R1, point 5):
The Coordinated Action Checklist (CAC), Appendix A, shows how the team the Connectors values the relations in the team and towards their other collaborative partners. The scores show the mean score of the individual ratings for each subject from 1-100. In itself, these evaluations are not valid measurements of the underlying concepts [36]. However, they are the starting point of the joint evaluation of the collaborations in the network, since the CAC is intended as a conversation starter [36]. The second group session (see Figure 1) was dedicated to the discussion following the CAC. We paid attention to the themes received high scores (i.e. the growth dimension and the suitability of partners), to learn why these themes have a higher score than the other themes and how to maintain this. In addition, we dwelt on the other themes and what it would take to score them higher. Individual questions from the checklist that scored remarkably low were likewise highlighted. For example, the fourth question “The contribution of the different members is to everyone’s full satisfaction” received low scores compared to other questions. It appeared that contributions from team members were not always acknowledged by other team members. This was also evident from results obtained from the individual evaluation interviews and other group sessions. The discussion showed us that for them, the contribution statement is closely aligned with the 11th statement: “The partnership functions well (working structure, working methods)” and the 15th statement: “The partnership partners work together in a constructive manner and know how to involve each other when action is needed”. Since the team started without a clear guideline and figured out themselves what works and what does not, it appeared that the team members had different ideas of what contributes to the team goals and what not. The individual evaluation interviews showed similar findings. Due to insufficient process evaluation in the first year (statement 12), which was also a topic of discussion in the first group session, this group session revealed that agreement on the goals of the team (statement 9) has to be discussed and improved. This was followed up in the following weeks and observed in our fieldwork during that period. From this discussion about the CAC statements, the team agreed to improve communication and express expectations to each other. Besides the reoccurring reflection sessions started as a result of the second group session about the CAC. The questionnaire was not repeated at a later time, but the topics of discussion were perceived as a valuable addition to the team’s evaluation and learning process and did return in the various reflection sessions. For this reason, the topics of the CAC are also integrated in other data sources.
Comment 4: The CAC scoring interpretation is not adequately explained. What do the different scores represent, and how should readers understand their significance?
Response 4: Thank you for this feedback. We have addressed the CAC scoring interpretation in the additional paragraph on p. 8, row 342-374 (See response to the comment above).
Regarding References:
Comment 1: The authors could update their references as only 13 of 40 are from the last 5 years (and some from municipal/national reports).
Response 1: Thank you for this observation. We initially used key publications in particular complemented by recent insights, which might explain why there are older references among them. We have updated our references to more recent years:
20. Schot, E., Tummers, L., & Noordegraaf, M. (2020). Working on working together. A systematic review on how healthcare professionals contribute to interprofessional collaboration. Journal of interprofessional care, 34(3), 332-342.
24. Dougall, I., Vasiljevic, M., Wright, J. D., & Weick, M. (2024). How, when, and why is social class linked to mental health and wellbeing? A systematic meta-review. Social Science & Medicine, 343, 116542.
35. Akyıldız, S. T., & Ahmed, K. H. (2021). An overview of qualitative research and focus group discussion. International Journal of Academic Research in Education, 7(1), 1-15.
42. Stephens, W., van Steden, R., & Schoonmade, L. (2024). Boundary spanning in local governance: A scoping review. Administration & Society, 56(2), 99-144.
Reviewer 3 Report
Comments and Suggestions for Authors
This is a well written, clearly structured, interesting article in an area of growing importance and attention. My only comment would be to perhaps define intersectoral as it is used in this context early on in the article e.g., to examine the interrelations between co-producers and professionals and to distinguish this usage from intersectoral distinctions of class, gender, ethnicity etc within the coproducers/ neighbourhood.
Author Response
Comment 1: This is a well written, clearly structured, interesting article in an area of growing importance and attention. My only comment would be to perhaps define intersectoral as it is used in this context early on in the article e.g., to examine the interrelations between co-producers and professionals and to distinguish this usage from intersectoral distinctions of class, gender, ethnicity etc within the coproducers/ neighbourhood.
Response 1: Thank you for the compliments and feedback. In this article, intersectoral does not refer to distinctions such as class, gender or ethnicity, but rather to the relationship between coproducers and professionals from various sectors, as outlined in the first paragraph of the introduction. To address your comment and clarify this in the text, we have added the following sentence just before the study objective: “Therefore, we define intersectoral collaboration as the collaboration between both professionals from diverse sectors and coproducers.” (p. 2, row 71-72)
Round 2
Reviewer 2 Report
Comments and Suggestions for Authors
Dear Authors,
Thank you for sending the revised version. I'm pleased that you acknowledged the comments and made improvements in quality and readability.
I wish you all the best in your future endeavours and all your work. Keep going with this area, it's exciting!
Best regards.